# Comparison of the Effects of High-Power Diode Laser and Electrocautery for Lingual Frenectomy in Infants: A Blinded Randomized Controlled Clinical Trial

**DOI:** 10.3390/jcm11133783

**Published:** 2022-06-30

**Authors:** Adriana Mazzoni, Ricardo Scarparo Navarro, Kristianne Porta Santos Fernandes, Raquel Agnelli Mesquita-Ferrari, Anna Carolina Ratto Tempestini Horliana, Tamiris Silva, Elaine Marcílio Santos, Ana Paula Taboada Sobral, Aldo Brugnera Júnior, Samir Nammour, Lara Jansinski Motta, Sandra Kalil Bussadori

**Affiliations:** 1Postgraduate Program in Biophotonics Applied to Health Sciences, Nove de Julho University (UNINOVE), São Paulo 01525-000, Brazil; adrianamazzoni@uol.com.br (A.M.); kristianneporta@gmail.com (K.P.S.F.); annacrth@gmail.com (A.C.R.T.H.); larajmotta@terra.com.br (L.J.M.); 2Postgraduate Program in Bioengineering, Scientific and Technological Institute, Brazil University (UB), São Paulo 08230-030, Brazil; ricardosnavarro@gmail.com; 3Postgraduate Program in Rehabilitation Sciences, Nove de Julho University (UNINOVE), São Paulo 01525-000, Brazil; raquel.mesquita@gmail.com (R.A.M.-F.); tamiris.slv@hotmail.com (T.S.); 4Postgraduate Program in Health and Environment, Metropolitan University of Santos (UNIMES), Santos 11045-002, Brazil; elaine.marcilio@unimes.br (E.M.S.); anapaula@taboada.com.br (A.P.T.S.); 5National Institute of Science and Technology, INCT “Basic Optics Applied to Life Sciences”, UFSCar, USP, São Carlos 13565-905, Brazil; aldobrugnera@gmail.com; 6Laser Application in Dental Medicine, Department of Dental Sciences, Faculty of Medicine, University of Liege, 4000 Liege, Belgium; nammourdent@hotmail.com

**Keywords:** ankyloglossia, frenectomy, laser, electrocautery, breastfeeding

## Abstract

The aim of the study was to evaluate the release of the lingual frenulum through frenectomy in newborns zero to 90 days of age who breastfed and had diagnosis of ankyloglossia with an indication for surgery, comparing two methods: electrocautery and a high-power diode laser. Fifty-seven patients were randomly allocated to two groups (23 submitted to electrocautery and 34 submitted to a high power diode laser). Tongue movements were evaluated based on a clinical assessment and using the Bristol Tongue Assessment Tool (BTAT) before and 15 days after the surgical procedures. The visual analog scale was administered to the mothers on the same occasions for the measurement of pain during breastfeeding. Both groups had an increased BTAT score (favorable outcome) at the post-surgical evaluation, but the anterior third of the tongue was not always free to enable the movements necessary for lingual functions. It is fundamental for surgeons to have skill and in-depth knowledge of the equipment used to avoid accidents and complications in the region of important structures. Both techniques employed in this study were safe and effective, causing little bleeding and few postoperative complications. The group submitted to a high-power diode laser exhibited less post-surgical bleeding compared to the group submitted to electrocautery and no inflammation at the edges of the surgical cut.

## 1. Introduction

Ankyloglossia (tongue-tie) is a congenital oral abnormality characterized by a short, anteriorized lingual frenulum, resulting in altered tongue movements, exerting an impact on functions such as sucking, speech, and eating [1]. This condition occurs in 4 to 16% of newborns and is more frequent in the male sex [2]. Ankyloglossia in early childhood is related to difficulties breastfeeding as well as air intake during breastfeeding and the introduction of foods. The condition can also cause delayed orofacial growth and development as well as speech and behavioral problems [3].

Lingual frenectomy can improve tongue mobility and can be performed using different techniques and surgical instruments, with variations in both operating and recovery times. This procedure can be performed with the conventional, cold blade instruments as well as thermal resources [4,5,6]. Thermal techniques include electrocautery, laser, and radiofrequency ablation [7]. Electrosurgery and laser frenectomy have advantages over more invasive techniques with cold blade instruments, such as less intraoperative and postoperative bleeding [4]. 

The aim of the present study was to evaluate the use of thermal cutting instruments for lingual frenectomy as well as direct impacts on improving tongue movements and breastfeeding within 15 days after the surgical procedure. Two thermal instruments were compared: surgical cuts performed with a high-power diode laser and electrocautery.

## 2. Materials and Methods

### 2.1. Study Setting 

The present study was conducted at a dental clinic. Due to the COVID-19 pandemic, the clinics of Nove de Julho University (UNINOVE) were closed to the public. The dental clinic at which the participants were examined followed all safety standards and protocols recommended by the Brazilian Health Ministry and government agencies for controlling the dissemination of the coronavirus. This study received approval from the local Ethics Committee (certificate number: 4.387.769) and received funding from the Coordination for the Advancement of Higher Education Personnel (CAPES) (number: 88887.494956/2020-00). The Informed Consent Term was explained to the parents or guardians of the newborn participants and signed in two copies—one belonging to the volunteer and the other to the researchers. This study was registered with Clinical Trials: NCT04487418

### 2.2. Design and Participants

A randomized, controlled, blind, clinical trial was conducted with 57 newborns zero to 90 days of age who were breastfed (exclusive or mixed). Recruitment took place from September 2020 to November 2021. All participants were within the standards of normality in terms of health and had a diagnosis of ankyloglossia with an indication for lingual frenectomy. The instrument used to assess the lingual frenulum was the Bristol Tongue Assessment Tool (BTAT), which is recommended by the Brazilian Health Ministry and was adjusted with the assessment of breastfeeding according to the recommendations of UNICEF. On this test, a score closer to 0 (zero) denotes greater severity with a possible indication for surgery, whereas a score closer to 8 (eight) is indicative of the normal pattern. A score of 0 to 4 could have surgical indication, depending on the breastfeeding evaluation. All breastfeeding was evaluated with a multidisciplinary approach. Infants with difficulties and a Bristol score of 0 to 4 points were indicated for surgery. A non-blinded researcher performed the assessments and treatments. 

Due to lack of relevant previous studies reporting the results of infants who underwent lingual frenectomy, it was not possible to calculate the sample size with known data. Therefore, the sample size calculation was performed with an estimate of the effect size, considering 0.25 as a minimal clinically relevant difference between the groups [8]. After the evaluation made by the operator researcher, a blinded examiner who had undergone training and calibration exercises performed the preoperative and postoperative evaluations. The classification of the type of frenulum was performed during the clinical evaluation: thin, submucosal, and thick. Among the 60 infants recruited, three were excluded from the sample for not meeting the eligibility criteria.

Allocation of the participants to the different groups was performed using a randomization model from the site www.randomizer.com (accessed on 24 July 2020). The participants were allocated to two treatment groups: (1) surgery with electrocautery and (2) surgery with a high-power diode laser (both instruments owned by the researcher).

### 2.3. Inclusion and Exclusion Criteria 

Inclusion criteria: Participants from the UNINOVE dental clinic in the city of São Paulo, Brazil, newborns zero to three months of age within the standards of normality in terms of health, no muscle or bone deviations, with exclusive or mixed breastfeeding, a diagnosis of ankyloglossia, and a score of 0 to 4 according to the Bristol protocol determined by the researcher in charge of the surgeries. 

Exclusion criteria: Newborns who were not being breastfed, those with congenital and systemic abnormalities, those with oral cavity abnormalities, those under medical treatment, those on medication, those who did not have good general health on the day of the surgical procedure and those who did not appear at the scheduled session.

### 2.4. Interventions 

Asepsis and infection control were rigorously followed for the preparation of the patients in accordance with biosafety norms. An analgesic (paracetamol) was administered in drops based on the weight of each participant 10 minutes prior to surgery for analgesia after the effect of anesthesia applied for the surgical procedure had worn off. 

Preoperative anesthesia was administered based on the study by Aras (2010), who indicated the use of a preoperative anesthetic prior to the high-power diode laser, which causes greater sensitivity compared to the Er:YAG laser. The anesthesia procedure involved the topical application of a pre-anesthetic ointment (20% benzocaine) using a suction device, followed by the topical application of anesthetic drops (tetracaine hydrochloride/phenylephrine) or infiltrative local anesthesia on the lingual frenulum using 3% mepivacaine. This anesthetic has fast action, which does not stress the newborn. A small amount is injected based on the patient’s weight with the needle, slowly and carefully in the region of lingual frenulum and the base of the tongue, depending on the patient age and condition of the lingual frenulum according to evaluation of the dental surgeon for each case [9,10].

The tongue was moved in the anteroposterior direction with grooved tongue elevator for better viewing. The cutting method described by Isac (2018) [11] was used. The surgical technique was performed by individualizing the frenulum with the aid of tweezers and the tongue elevator, followed by an incision starting at the free portion of the frenulum until reaching the base of the tongue. A longitudinal incision 2–3 mm in length was made in the frenulum, with the detachment of the fibers that limited tongue movement, and posteriorly until the wound had a rhomboidal shape [12]. The cut used for both surgeries was performed with sweeping movements with the tip at 45° to the fibers of the lingual frenulum with the aim of both executing the cut and achieving local clotting. Five to seven sweeping movements were performed with less than one second for each movement and paused to allow cooling, respecting signs of overheating at the cutting site.

Group 1—Surgery with electrocautery: Cut using electrocautery unit (BS Novadur Bayer/Cauterimax) with two basic waves that have two effects on tissues (cutting and clotting), both operating at the same power, maintaining constant cutting and cauterization [12]. Surgery with electrocautery is a safe, effective procedure. The parameters for electrocautery surgery are listed in Table 1.

Group 2—Surgery with high-power diode laser (Thera laser surgery; DMC, São Carlos, Brazil): Procedure performed with effective, safe energy that avoided thermal damage using controlled heating with the aim of cutting, clotting, promoting hemostasis, vaporizing soft tissue and fibers of lingual frenulum, achieving thermal photo effects and possible secondary photobiomodulation effects while minimizing the risk of harm to adjacent and deep structures. The parameters for high-power diode laser surgery are listed in Table 2.

Postoperative suturing was not necessary in either surgery. The use of postoperative anesthetic was prescribed based on the weight of the participant (paracetamol every six hours for three days after surgical procedure). Those responsible for the participants were previously informed about the possible complications of the surgery, as recommended by Isac (2018), such as hemorrhage, infection, damage to adjacent structures (muscles and sublingual caruncles), and fibrotic scarring leading to a risk of relapse [11].

Evaluations were conducted prior to the surgeries as well as at the follow-up appointment 15 days after the surgery, involving the administration of the Bristol protocol, standardized photography of the region of the lingual frenulum, and the administration of the visual analog scale (VAS) to the mother to measure pain during breastfeeding. This time was chosen based on studies by Azevedo (2005) [13], Trevisan (2010) [14], and Shah et al. (2013) [15] and because it is not possible to wait longer for an evaluation of breastfeeding infants due to the risk of early weaning. 

Because it is a controlled clinical trial and seeking greater transparency and high research standards, CONSORT (Consolidated Standards of Reporting Trials) recommendations, Figure 1, were followed.

### 2.5. Data Analysis 

Descriptive and inferential statistics were performed using the IBM SPSS statistical package, version 20. The Mann–Whitney test was used to determine differences in sex between groups as well as the Bristol results before and after surgery. 

## 3. Results

Fifty-seven newborns participated in the present study. Males predominated in the overall sample, with no significant differences between groups regarding sex. 

After surgery, relapse occurred in 26 cases (45.6%), requiring a second surgery. Recurrence was considered when the surgical site presented local stiffening, indicating inadequate wound contraction and fibrotic healing, or when the tissues were re-coaptated—both favoring a lack lingual movement previously achieved after the surgery. In such cases, the procedure needed to be redone to achieve subsequent success in moving the anterior region of the tongue, which was our main objective. Although increases in the Bristol score were found at the 15-day evaluation after lingual frenectomy for most patients in both groups, some cases presented recurrence during this period, with the need to redo the surgical procedure. Surgical difficulties were encountered irrespective of the instrument used, with thicker and submucosal frenula posing greater cutting difficulty. The distribution of the types of frenula in the groups is displayed in Table 3. A thin frenulum predominated, and a low frequency of submucosal frenulum was found in both groups.

Table 4 displays the results of behavior after surgery (crying and bleeding) according to the two surgical procedures. A significant difference between groups was found with regards to bleeding, which was more frequent in the electrocautery group.

No significant differences were found between evaluation times (before and after surgery) regarding symptoms related to breastfeeding, such as gas, hiccup, colic, clicking sounds, and burping. The hypothesis was an improvement in the Bristol score after surgery, as the condition that limited tongue movements was reduced by the procedures. However, despite finding similar postoperative improvement in lingual movements in both groups, some cases presented recurrence (Figure 2 and Figure 3).

A better result in terms of mother’s pain during breastfeeding at the postoperative evaluation was found for the electrocautery group (Figure 4). However, factors independent of surgery may be related to this finding. 

Local inflammation at the edges of the surgical cut was found in four cases in the electrocautery group and none in the laser group. After the surgical procedure, 26 of the 57 participants relapsed; 6 (26.6%) in the electrocautery group and 20 (58.8%) in the laser group). The researchers considered the following conditions indicative of relapse: limited or no tongue movement, hardened fibrotic tissue at surgical cutting site perceived through clinical viewing and palpation, return of anteriorization of lingual frenulum, lack of tongue mobility assessed clinically and using the Bristol protocol, inefficient breastfeeding in post-surgical period, and need to perform surgery a second time to release the anterior third of the tongue.

## 4. Discussion

Considering the difficulty that ankyloglossia can pose for breastfeeding, the present study compared the effects of two different thermal methods (high-power diode laser and electrocautery) for lingual frenectomy. The findings confirm that lingual frenectomy leads to an improvement in tongue mobility but it is essential for the surgeon to choose a safe, effective method for the patient and have both skill and knowledge regarding the protocols for the use of these instruments, the surgical cutting technique to detach the fibers of the frenulum that limit the movements of the tongue, and the expected effects in the postoperative evaluation. This surgery can be performed with a cold-blade scalpel, but this type of instrument promotes greater bleeding compared with thermal instruments, such as electrocautery and a high-power diode laser [1,7,16,17,18].

In all cases, it is necessary to remove the part of the lingual frenulum that limits the anterior region of the tongue. Thus, the detachment of the fibers that are causing the limitation of tongue movements is fundamental. These fibers are attached anteriorly on the underside of the tongue and/or the floor of the oral cavity or even submucosally and with a more posterior attachment. It is important for the procedure to be executed with the aim of achieving success in a single session, but relapse can occur, with the need for additional surgery, as occurred in some of the cases in the present investigation [19,20].

According to Mello Neto (2021), early diagnosis and lingual frenectomy promote the reestablishment of the functions of the stomatognathic system as well as the organic movements of the tongue in the period subsequent to surgery. Postoperative follow-up requires multidisciplinary interventions by specialists in speech therapy or oral physiotherapy, with the careful observation of possible relapse and the need to redo the surgical procedure [21].

Messner et al. (2020) published a clinical consensus of the American Academy of Pediatric Otolaryngologists based on clinical statements developed by a panel of experts using an objective research method. The group found marked differences regarding the diagnosis of ankyloglossia and the need for frenotomy in infants to promote improved breastfeeding. It has been described that not all infants with ankyloglossia need to undergo a frenotomy and there are other causes that limit breastfeeding. There is limited evidence supporting frenectomy for the treatment of children with phonation problems. More studies are needed to refine the selection of the most indicated patients and treatments. There is a lack consensus, gaps in knowledge, and a lack of evidence on the diagnosis, management, and treatment of ankyloglossia [22].

The choice of surgical laser in the present investigation was based on the study by Amaral et al. (2015), who report advantages such as hemostasis, faster healing, and the better formation of new tissue as well as reductions in postoperative pain and the use of local anesthesia [23]. However, the reduction in pain when using a high-power diode laser has been questioned. Aras et al. (2019) compared tolerance to lingual frenectomy based on the need for local anesthesia and postoperative pain on the part of patients having undergone surgery with diode laser or erbium: yttrium-aluminum-garnet (Er: YAG) laser and indicated that only Er: YAG laser can be used for lingual frenectomy without local anesthesia. In the present study, the high-power diode laser was used with the same anesthesia protocol as that used for electrocautery [24].

The hypothesis of the present investigation was based on studies by Oliveira et al. (2010) and Nunes et al. (2021), in which lingual frenectomy was performed with lasers, as the increase in temperature decontaminates the site and favors tissue repair without the occurrence of infection, enabling greater efficiency and patient comfort compared to traditional methods with a cold-blade scalpel [17,25]. However, postoperative fibrotic scarring or re-coaptation of tissues that have been cut occurred in a larger part of the patients submitted to the high-power laser in comparison to electrocautery, leading to relapse. 

A probable explanation for the formation of fibrotic scar tissue in surgeries with the high-power diode laser is that, besides the high temperature emitted at the surgical site, the use of the high power diode laser, near the infrared wavelength within the therapeutic window of photobiomodulation, used with careful technique and controlled parameters, aiming to minimize thermal damage, promotes a faster healing process due to the increase in local blood circulation, favoring the contraction of the scar, and the lowered tongue position due to the lingual frenulum favoring the re-coaptation of the tissues, increasing the likelihood of the formation of fibrotic scar tissue, with the loss of tongue mobility and the occurrence of relapse [26]. These results contradict findings described in the study conducted by Koppolu (2016) [27].

Thus, there is no consensus or evidence in the literature to indicate a specific surgical protocol for newborns due to the scarcity of scientific production on this subject [17] since newborns who are breastfed cannot wait for a long time for the tongue to heal and experience the release of lingual movements. Thus, a procedure that has a quick postoperative effect on lingual movements is necessary to avoid early weaning. The definition of modulation parameters and the establishment of laser protocols are of considerable importance for this population. Indeed, Fukuda and Malfatti (2008) draw attention to the need for methodological standardization in clinical trials involving laser, as the divergent results reported in the literature call into question the benefits of this method [28].

According to Sant’Anna et al. (2017) and Charoenkwan et al. (2017), the laser power configuration should be adjusted to the lowest necessary intensity to avoid damaging the target tissue. In the present study, the power was approximately 1.5 W [29,30]. However, more in-depth studies are needed to determine the parameters for a high-power laser in surgeries conducted with newborns to establish a protocol that considers the needs of this age group, as infants have a more accelerated metabolism and increased blood circulation compared to adults, which favors rapid healing, thereby facilitating the formation of fibrotic scar tissue [31].

The present results are in disagreement with findings described by Shah et al. (2013), who concluded that all surgical instruments generate a similar healing process. In contrast, a significant difference was found in the present study with regards to postoperative bleeding (within the expected pattern of normality) and local inflammation at the edges of the surgical cut, which was more frequent in the group submitted to electrocautery, indicating a difference in the effects of the procedure compared to the high-power diode laser in terms of recovery [15].

The high-power laser used for the removal of soft tissue in contact mode promotes a similar tactile and clinical sensation as that found with electrocautery. However, the authors of the present study believe that the use of the high-power diode laser could have had a better postoperative outcome if some of the parameters were changed, as Ghagheri (2017 study) had a more favorable result using the high-power diode laser at the wavelength of 1.064 nm in pulsed mode. In a qualitative analysis, the present results are in agreement with findings described by Araujo et al. (2007) and Andrade et al. (2007), who demonstrated a greater occurrence of fibrotic scar tissue in surgeries performed with a high-power diode laser due to the fact that the laser light promotes an increase in fibroblastic proliferation and the acceleration of epithelialization in wounds, even if this is a secondary effect [19,20,32].

There is a need for the clear identification of whether tongue movement difficulties remain in the occurrence of the re-coaptation of the tissues or a fibrotic postoperative healing process. Indeed, this could be a mistake in the results of previous studies with regards to healing. It is important to observe whether the tissues in the region underwent a new coaptation because they were close or whether the local tissue is whitish and hardened at the end of the complete healing process, as fibrotic scar tissue could favor a reduction in tongue mobility, leading to possible relapse. In addition to being an environment with saliva and that contains histatin-1, which is responsible for accelerating healing, the participants of this study had an accelerated metabolism and greater blood circulation compared to adults and, therefore, the parameters for this age group must be very different [14,33,34].

The follow-up evaluation in the present study was limited to 15 days after the procedure because the participants were newborns who need to have their lingual movements released to be able to breastfeed effectively. Future studies should involve a longer follow-up period to investigate the possibility of the appearance of fibrotic scar tissue in tissues adjacent to the cut or re-coaptation of nearby local tissues in a period longer than two weeks, which may limit the free movements of the tongue [13,15].

Although differences were found in the BTAT score between examiners, this did not imply a change in the diagnosis or treatment plan for each case. The authors agree that there is no gold-standard protocol for the evaluation and a multidisciplinary oral evaluation is needed for infants so that the diagnosis and treatment of ankyloglossia can be more assertive.

Regarding the breastfeeding assessment, tongue movements and some aspects of breastfeeding were improved in most patients. However, the desired breastfeeding success was not achieved in all cases in both groups. A more in-depth analysis of this issue is needed. Besides investigating further options in terms of surgical methods, postoperative follow-up with a speech therapist and physiotherapist is needed to achieve successful postoperative breastfeeding, with reassessment and adjustments of the breastfeeding practice. The infant needs to learn new latching and sucking techniques and the orofacial musculature needs to be worked due to its altered development resulting from the limited tongue movements prior to surgery.

It should be pointed out that conservative, minimally invasive options exist that involve tongue movement therapy to avoid the re-coaptation of tissues and possible relapse, as well as establish adequate tongue movements without limitations to proper sucking or swallowing. However, minimally invasive surgery is possible with thermal instruments, such as high-power laser and electrocautery.

The present findings underscore the need for further studies to define the type of surgical cut made with thermal instruments so that the heat is not dissipated to adjacent tissues and to determine the power required for each case according to patient age and the clinical condition of the lingual frenulum. Future studies should investigate the most suitable parameters for the use of a high-power diode laser for newborns [30,31].

Future clinical and animal model studies should be carried out to compare lingual frenectomy surgery with the traditional method and minimally invasive methods, with different high-power laser wavelengths, irradiation parameters (irradiance), cooling methods, trans-surgical movement, whether or not to perform photobiomodulation therapy in the postoperative period, aiming to evaluate the healing patterns in the postoperative period and the most suitable parameters for the use of a high-power laser for newborns in the treatment of ankyloglossia.

## 5. Conclusions

After the statistical analysis, the authors realized that the methods used for the treatment of ankyloglossia proved safe and well tolerated, with no significant difference between the two techniques. However, post-surgical inflammation of the cut edges was less frequent in the group submitted to the high-power diode laser with the parameters used in this study compared to the electrocautery group. Little clinical difference was found between the two instruments, but the use of the high-power diode laser led to more cases of postoperative of relapses (58.8%) when compared to the use of electrocautery. 

## Figures and Tables

**Figure 1 jcm-11-03783-f001:**
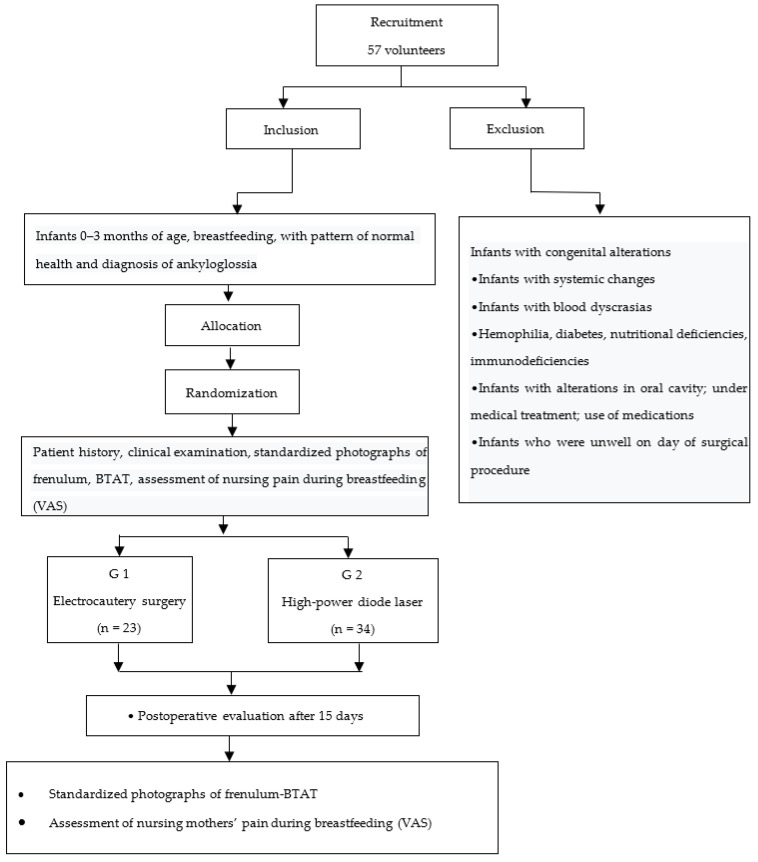
Clinical study flowchart.

**Figure 2 jcm-11-03783-f002:**
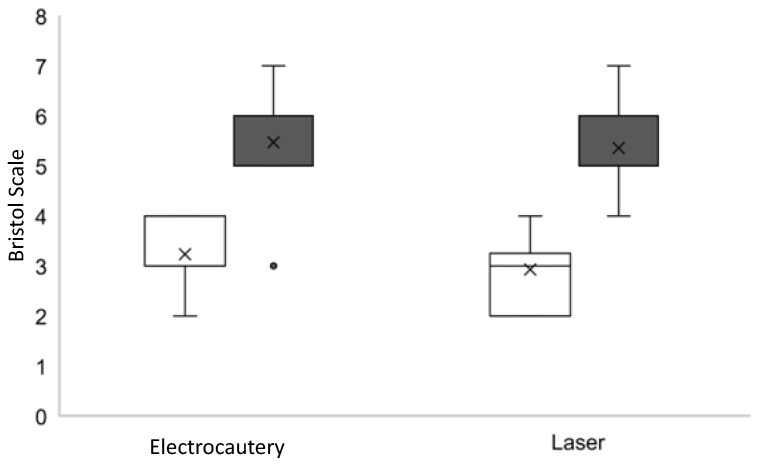
Assessment of ankyloglossia based on Bristol protocol (researcher 1). White = Before. Black = After.

**Figure 3 jcm-11-03783-f003:**
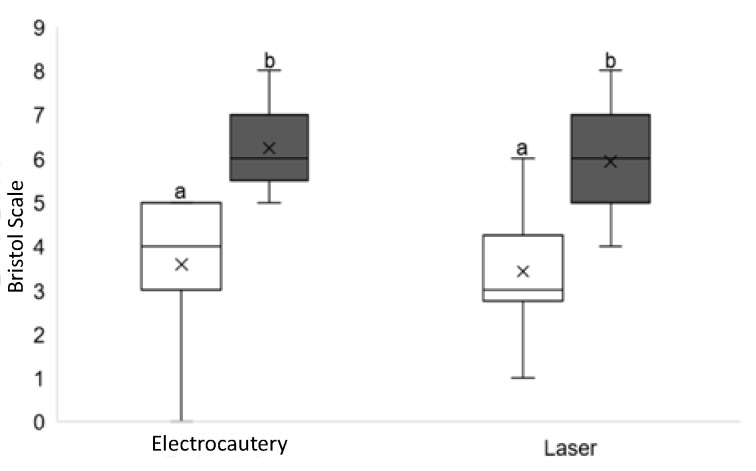
Assessment of ankyloglossia based on Bristol protocol (researcher 2). White = Before. Black = After.

**Figure 4 jcm-11-03783-f004:**
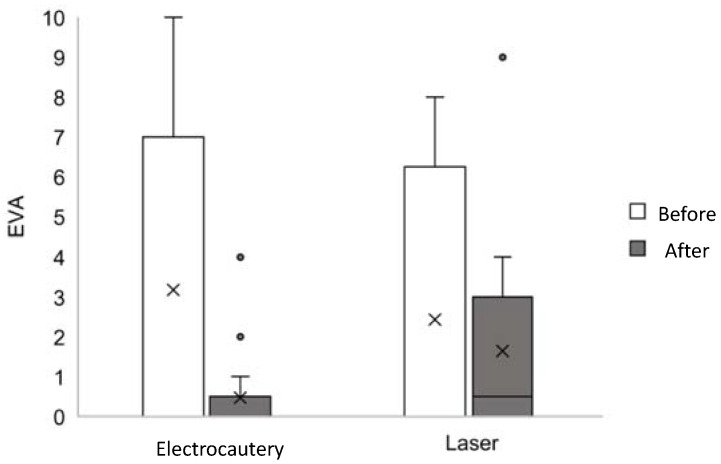
Visual analogue scale for mother’s pain during breastfeeding.

**Table 1 jcm-11-03783-t001:** Parameters for electrocautery.

Electrocautery Device (BS Novadur Bayer/Cauterimax)
Fine tip with nickel-chromium filament
Filament length: 18 cm
Power only: 1200 °C
Operating mode: continuous by contact with target tissue
Area: approximately 1 cm

**Table 2 jcm-11-03783-t002:** Parameters for the high-power diode laser.

High-Power Diode Laser (DMC: Thera Laser Surgery)	
Wavelength	808 nm ± 10 nm (infrared)
Maximum power of the equipment	Up to 9 W ± 20%
Operating mode	Continuous (CW); exposure limited by pedal switch
Used Power	1.5 to 2 W
Area	Approximately 1 cm
Irradiation mode	Contact with target tissue
Useful diameter of optical fiber	600 micrometers

**Table 3 jcm-11-03783-t003:** Distribution of types of frenulum in different groups.

Group	Type of Frenulum	n
Electrocautery	Thin	11 (47.8%)
	Submucosal	5 (21.6%)
	Thick	7 (30.4%)
	Total	23
Laser	Thin	18 (52.9%)
	Thick	8 (23.5%)
	Submucosal	8 (23.4%)
	Total	34

**Table 4 jcm-11-03783-t004:** Postoperative observation.

	Group		N	%	*p **
Crying	Electrocautery (n = 23)	Yes	23	100%	0.241
		No	0	0	
	Laser (n = 34)	Yes	32	94.1%	
		No	2	5.9%	
Bleeding	Electrocautery (n = 23)	Yes	10	43.5%	0.006
		No	13	56.5%	
	Laser (n = 34)	Yes	27	79.4%	
		No	7	20.6%	

* Mann-Whitney test.

## Data Availability

The datasets (Excel spreadsheets) generated from this protocol are available from the corresponding author (Sandra Kalil Bussadori—sandra.skb@gmail.com) upon any reasonable request. However, reuse of this data will not be permitted for anyone who is not an author of this paper.

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
