# Peer review of "Comparison of the Effects of High-Power Diode Laser and Electrocautery for Lingual Frenectomy in Infants: A Blinded Randomized Controlled Clinical Trial"

_jcm, 2022, doi:10.3390/jcm11133783_

Round 1

Reviewer 1 Report

The theme of the paper is original and interesting.
The authors presented the aims of their research.

Material and method are clearly presented.

The paper addressed the proposed subject.
The conclusions are consistent through the presented arguments.

22 bibliographic titles of 36 are older than 5 years.

Reviewer 2 Report

The paper "Comparison of the effects of high-power diode laser and electrocautery for lingual frenectomy in infants: a blinded randomized controlled clinical trial" is based on a small sample of patients, 17 in the electrocautery group, 14 in the laser group. Follow-up is definitely too short to properly assess the quality of scar and to compare the results obtained by either electrocautery or laser surgery.  To get sound results, a third group of patients, treated by surgery, should be added and longer observation of both scar and infants' functions should be carried on.  Complications, early and late, observed after performed procedures are very important and should be presented in detail. Bleeding is not a "behavior" but a serious complication, especially in newborns.

At the moment this study does not include sufficient data to compare the results and the role of laser and electrocautery in lingual frenectomy in infants.  

Reviewer 3 Report

The authors conducted a single-blind RCT in newborns with ankyloglossia to compare high-power laser lingual frenotomy to electrocautery and found that less postoperative bleeding and inflammation occurred in the group who received laser.

The abstract is in line with the instruction for the authors and gives the impression that the scores of the Bristol Tongue Assessment Tool and Visual Analogical Scale for nipple pain during breastfeeding may be the main outcomes of the study but no results regarding those outcomes are reported. Safety, effectiveness, bleeding, and postoperative complications seem secondary outcomes of the study but those also lack a clear definition. The sentence in line 31, “Both techniques employed in this study were safe and effective, causing little bleeding and few postoperative complications”, is a duplicate of the sentence in line 29. Revisions may improve the readability and quality of the abstract.

The introduction section is brief and provides what the reader needs to know about ankyloglossia and frenotomy. Moreover, the aim of the study has a different connotation from the one present in the abstract (“evaluate whether the choice of cutting instrument […] exerts a direct impact on improvements in tongue movement and breastfeeding” VS “evaluate the release of the lingual frenulum […] comparing the use of electrocautery and high-power laser”). The authors should choose the connotation that best fits the aim of the study and report in both sections to improve the consistency of the manuscript.

The Materials and Methods section lacks essential information, such as a statement regarding the compliance with the Helsinki Declaration, the approval by the local Ethical Committee or Institutional Review Board, as well as any reference to informed consent provided by parents or legal guardians of the newborns who were enrolled in the study. In addition, the authors don’t provide information regarding any registration of the RCT. In this section, the authors anticipate that they enrolled sixty newborns during the study period, but they don’t describe any sample size calculation therefore the reader cannot know if the N of the study is sufficient to obtain significant results. The authors used the Bristol Tongue Assessment Tool to diagnose ankyloglossia without explaining this tool: the addition of a table that provides a full explanation of such Bristol protocol could improve the reading experience of those who don’t know such a diagnostic system. In lines 94 to 96, the authors describe unclearly their protocol for local anesthesia: did they use benzocaine 20% alone or associated with tetracaine/phenylephrine drops? Moreover, it is important to point out that infiltrative anesthesia is not recommended in newborns; to deepen such feature, it is useful to read (and cite) the paper by Messner et al. (Messner AH, Walsh J, Rosenfeld RM, et al. Clinical Consensus Statement: Ankyloglossia in Children. Otolaryngol Head Neck Surg. 2020;162(5):597-611. doi:10.1177/0194599820915457). The electrocautery parameters described in line 109 are repeated in line 112; the creation of a table also explaining those parameters (as the one available for laser) could improve the readability of the manuscript. In table 1, the authors describe the parameters of their high-power diode laser distinguishing unclearly “useful power = up to 9 W ± 20%” from “power = 1.5 to 2 W”. Several studies showed that diode laser performs efficiently by using a power of 7 W (in continuous wave mode); to deepen, it is useful to read (and cite) the paper by Limongelli et al. (Limongelli L, Capodiferro S, Tempesta A, et al. Early tongue carcinomas (clinical stage I and II): echo-guided three-dimensional diode laser mini-invasive surgery with evaluation of histological prognostic parameters. A study of 85 cases with prolonged follow-up. Lasers Med Sci. 2020;35(3):751-758. doi:10.1007/s10103-019-02932-z). The latter could explain the high percentage of patients who needed reintervention in the laser group. In the “Data analysis” section, the authors report that used the Mann-Whitney test to determine differences in sex, but this is incorrect because such a test fits with non-parametric ordinal variables and sex is a qualitative dichotomic one. The authors should clearly state which are the main outcomes and secondary outcomes of the comparison between electrocautery and high-power laser, then should state which statistical tests were used to perform such comparisons, and eventually should set the level of evidence for each statistical test. The material and methods section requires extensive review. It is strongly recommended to follow the principles of the Consolidated Standards of Reporting Trials (CONSORT) statement, available at http://www.consort-statement.org/.

The results section is poor because of the imperfections of the previous section and because of other intrinsic weaknesses. Table 2 shows a classification of the type of frenulum that is not mentioned in the methods. The total participants of the electrocautery group are 17 patients in table 2, 19 patients in the upper part of table 3, 16 patients in the lower part of table 3, and 23 in the abstract; the total of the high-power laser group is 14 patients in table 2, 25 in the lower part of table 3, 30 in the upper part of table 3, and 34 in the abstract. Why the number of participants is so unstable? Figures should be associated with brief dedicated explanations of the contents and abbreviations. The phrase from lines 176 to 181 should belong to the methods section because explains how the authors established the need for reintervention. The results need massive revisions and explanations in line with the material and methods section.

Discussion is appreciable and could be further improved by adding the above-mentioned papers:

·       Messner AH, Walsh J, Rosenfeld RM, et al. Clinical Consensus Statement: Ankyloglossia in Children. Otolaryngol Head Neck Surg. 2020;162(5):597-611. doi:10.1177/0194599820915457

·       Limongelli L, Capodiferro S, Tempesta A, et al. Early tongue carcinomas (clinical stage I and II): echo-guided three-dimensional diode laser mini-invasive surgery with evaluation of histological prognostic parameters. A study of 85 cases with prolonged follow-up. Lasers Med Sci. 2020;35(3):751-758. doi:10.1007/s10103-019-02932-z.

Conclusions need to be revised after the revisions in the results section of the manuscript.

In my opinion, the manuscript could be interesting but requires extensive revisions to be accepted. I suggest reconsidering the manuscript after MAJOR REVISIONS.

Round 2

Reviewer 2 Report

Thank you for the corrections and explanations. Yet, I would advise the title of table 4: "Postoperative observation" instead of "Behavior in different groups after surgery". Bleeding is definitely not "a behavior". In section 5 "Conclusions" a very high relapse rate (58,8%) in the laser group should be stressed.
I strongly recommend comparing, in the future, results of traditional surgery with minimally invasive methods, which, as you have proven in your paper, lead to serious scarring and relapse in children with ankyloglossia .  

Author Response

Dear reviewer,  thank you for analyzing the article and raising important issues. We have revised and included the information as suggested. The Manuscript with track changes is attached.

Reviewer 3 Report

The authors successfully revised their manuscript, but further minor revisions are necessary before acceptance.  

The repetition of the word “with” (line 48) should be removed.

The sentence “Bristol protocol score of 0-4” (line 78) should be removed because is an anticipation of the following paragraph in which the authors explain clearly what the Bristol protocol score is.

Lines 87-88 and 92-94 are conflicting: “A non-blinded researcher performed the assessments and treatments.” VS “A blinded examiner who had undergone training and calibration exercises performed the preoperative and postoperative evaluations.”.

In table 2, the authors describe the parameters of their high-power diode laser distinguishing unclearly “useful power = up to 9 W ± 20%” from “power = 1.5 to 2 W”. Which power did they use to perform frenotomy?

Table 4 is a problem because the authors should recheck the math:

·       Crying in Electrocautery group, 19 “yes” + 0 “no” = 19 patients, not 23.

·       Crying in Laser group, 29 “yes” + 2 “no” = 31 patients, not 34.

·       Bleeding in Electrocautery group, 8 “yes” + 11 “no” = 19 patients, not 23.

·       Bleeding in Laser group, 24 “yes” + 7 “no” = 31 patients, not 34.

Figures 2 and 3 seem to display the same data. Why should the manuscript contain both?

In my opinion, the manuscript should be reconsidered after MINOR REVISIONS.

Author Response

(The authors gave the same response as above.)
